# Expression of SARS-Cov-2 Entry Factors in Patients with Chronic Hepatitis

**DOI:** 10.3390/v14112397

**Published:** 2022-10-29

**Authors:** Chiara Rosso, Cristina Demelas, Greta Agostini, Maria Lorena Abate, Marta Vernero, Gian Paolo Caviglia, Daphne D’Amato, Angelo Armandi, Marta Tapparo, Marta Guariglia, Giulia Troshina, Alessandro Massano, Antonella Olivero, Aurora Nicolosi, Antonella Zannetti, Rinaldo Pellicano, Alessia Ciancio, Giorgio Maria Saracco, Davide Giuseppe Ribaldone, Elisabetta Bugianesi, Sharmila Fagoonee

**Affiliations:** 1Department of Medical Sciences, University of Turin, 10126 Torino, Italy; 2Department of Molecular Biotechnology and Health Sciences, Molecular Biotechnology Center, University of Turin, 10126 Turin, Italy; 3Institute of Biostructure and Bioimaging, CNR, 80145 Naples, Italy; 4Gastroenterology Unit, Città della salute e della Scienza Hospital, 10121 Turin, Italy; 5Institute of Biostructure and Bioimaging, CNR c/o Molecular Biotechnology Centre, 10126 Turin, Italy

**Keywords:** ACE2, chronic hepatitis, NAFLD, SARS-Cov-2, TMPRSS2

## Abstract

Chronic hepatitis (CH) of dysmetabolic or viral etiology has been associated with poor prognosis in patients who experienced the severe acute respiratory coronavirus virus-2 (SARS-Cov-2) infection. We aimed to explore the impact of SARS-Cov-2 infection on disease severity in a group of patients with CH. Forty-two patients with CH of different etiology were enrolled (median age, 56 years; male gender, 59%). ACE2 and TMPRSS2 were measured in plasma samples of all patients by ELISA and in the liver tissue of a subgroup of 15 patients by Western blot. Overall, 13 patients (31%) experienced SARS-Cov-2 infection: 2/15 (15%) had CHB, 5/12 (39%) had CHC, and 6/15 (46%) had non-alcoholic fatty liver disease (NAFLD). Compared to viral CH patients, NAFLD subjects showed higher circulating ACE2 levels (*p* = 0.0019). Similarly, hepatic expression of ACE2 was higher in subjects who underwent SARS-Cov-2 infection compared to the counterpart, (3.24 ± 1.49 vs. 1.49 ± 1.32, *p* = 0.032). Conversely, hepatic TMPRSS2 was significantly lower in patients who experienced symptomatic COVID-19 disease compared to asymptomatic patients (*p* = 0.0038). Further studies are necessary to understand the impact of COVID-19 in patients with pre-existing liver diseases.

## 1. Introduction

The severe acute respiratory coronavirus virus 2 (SARS-Cov-2) pandemic has posed significant threats to global public health, not only due to acute respiratory distress syndrome, but also through multi-organ damage and dysfunction [1]. Several comorbidities, such as obesity, diabetes, and hypertension, have been associated with poor prognosis in patients with SARS-Cov-2 disease (COVID-19) and the synergism between chronic liver disease and SARS-Cov-2 infection may predispose to a worse prognosis [1]. Indeed, SARS-Cov-2 infection has been found to be associated with liver function abnormalities and disease severity [2,3,4,5,6,7].

SARS-Cov-2 engages, through the Spike (S) protein, angiotensin-converting enzyme 2 (ACE2) as the entry receptor and employs the cellular serine protease TMPRSS2 for S protein priming [8]. The role of ACE2 in liver disease is of special interest due to its regulatory activity on the renin-angiotensin system (RAS) that affects hepatic inflammation and tissue remodeling contributing to the pathogenesis of hepatic fibrosis [9,10,11]. The ACE2 ectodomain can be released in soluble form through the action of disintegrin or host proteases such as metalloproteinase 17 (ADAM17); the soluble ACE2 may also facilitate virus cell entry through receptor-mediated endocytosis [12].

Moreover, ACE2 can be transferred to other organs (such as the liver) through exosomes, a subpopulation of extracellular vesicles (EVs) that are released by all cell types, thus spreading SARS-Cov-2 infection [13,14,15]. Several studies have shown that, in the liver, both cholangiocytes and hepatocytes are susceptible to SARS-Cov-2 infection due to their ability to express ACE2, which increases significantly in the presence of pre-existing liver diseases [16,17,18]. However, the mechanism by which SARS-CoV-2 infection causes liver damage and affects the clinical outcome in patients with chronic hepatitis has not yet been fully investigated.

In this study, we aimed to explore the impact of SARS-Cov-2 infection with respect to the level of circulating and hepatic ACE2 and TMPRSS2 receptors as well as their associations with liver disease severity in a group of patients with chronic viral hepatitis of different etiologies.

## 2. Materials and Methods

### 2.1. Study Population

From 69 consecutive patients with chronic viral hepatitis of different etiology who underwent their clinical examinations during the COVID-19 pandemic between February 2020 and September 2020, 27 accepted to participate in the study. Similarly, from a group of 50 NAFLD patients who underwent liver biopsy before January 2019, 15 underwent their outpatient visit between February 2020 and September 2020 and were included in the study. All of the patients were enrolled at the Division of Gastroenterology and Hepatology of the University Hospital Città della Salute e della Scienza in Turin, Italy. A flow-chart of the study is depicted in Figure 1.

Overall, out of the 42 patients enrolled, 15 patients (35.7%) had chronic hepatitis B [CHB] infection, 12 (28.6%) had chronic hepatitis C [CHC] infection, and 15 (35.7%) suffered from non-alcoholic fatty liver diseases [NAFLD]. The inclusion criteria were: positive serology for CHB patients; detectable anti-HCV antibodies for CHC patients; diagnosis of NAFLD confirmed by liver biopsy in patients without alcohol consumption according to established thresholds (less than 210 g/week for males and 140 g/week for females). Patients with other etiologies of chronic hepatitis, such as autoimmune hepatitis, primary biliary cirrhosis, alcoholic liver disease, and hemochromatosis were excluded. SARS-Cov-2 infection was based on quantitative reverse transcription-polymerase chain reaction or positive serology. All the personnel performing the experiments were blinded to the clinical characteristics of the patients. The local Ethics Committee approved the study protocol (CEI/522, 17 November 2015); all the patients enrolled provided written informed consent for participation. The study conformed to the principles of the Helsinki Declaration.

### 2.2. Liver Biopsies

Liver biopsies were examined by a single expert liver pathologist who was blinded to the patients’ clinical characteristics. The average size of the liver biopsies was 25 mm (range 14–45), and they had a minimum of 11 portal tracts; inadequate biopsies were excluded. Kleiner classification was used to establish the diagnosis of NASH, according to the joint presence of steatosis, hepatocyte ballooning, and lobular inflammation with or without fibrosis [19].

### 2.3. Western Blot Analysis

A total of 15 liver biopsies were available for protein extraction. Biopsies had been collected in RNAlater (Ambion Inc., Austin, TX, USA) and stored at −80 °C until processing. Liver tissues were homogenized by Tissue Lyzer and lysate for protein extraction was obtained using the *mir*Vana^TM^ PARIS^TM^ RNA according to the manufacturer’s instructions, followed by storage at −80 °C.

To the thawed homogenate, a cocktail of protease inhibitors (Complete Mini, Roche, Monza, Italy) was added. Twenty μg of protein samples was subjected to SDS-PAGE using Mini-PROTEAN Precast gels (4–15%) (Bio-Rad, Hercules, CA, USA) and processed as previously described [20]. Protein quality was assessed by Ponceau S (Ponceau S stain, 0.01% (*w*/*v*) in 1% acetic acid (*v*/*v*)) staining of membrane. The following primary antibodies were employed: rabbit anti-human ACE2 (ab108252) at 1:1000 dilution in 1% BSA in TBS-Tween, rabbit anti-human TMPRSS2 (ab109131, Abcam, Cambridge, UK) at 1:1000 dilution and mouse anti-vinculin primary antibodies at 1:8000 dilution (in-house) [20]. Goat anti-rabbit and anti-mouse HRP-conjugated secondary antibodies (Sigma) at 1:5000 dilution were used followed by enhanced chemiluminescence signal detection by the Chemidoc system (Bio-Rad, Hercules, CA, USA). A densitometric analysis was performed using Image Lab 6.1 software (Biorad Laboratories, Segrate, Italy). Protein extracts were obtained from Caco-2 intestinal cells, as previously described, as they are known to express high quantities of ACE2 and TMPRSS2 and can be used as a positive control [20].

### 2.4. Circulating ACE2 and TMPRSS2 Determinations

Blood samples were collected at the time of the outpatient visit (for CHB and CHC patients) or at the time of liver biopsy (for NAFLD patients) and stored at −80 °C until analysis. Plasma levels of ACE2 and TMPRSS2 were measured by the commercially available human enzyme linked immunosorbent assay (ELISA) kit (Abcam, Cambridge, UK), according to the manufacturer’s instructions. The concentration of ACE and TMPRSS2 was determined with an ELISA reader at 450 nm. The intra- and inter-assay coefficients of variation were below 5% for ACE2 and below 6% for TMPRSS2. The concentration of both targets was expressed as nanograms per milliliter (ng/mL).

### 2.5. EV Isolation and Characterization, Western Blotting and ELISA

EVs were isolated from 200 uL of serum previously collected and stored at −80 °C, using ExoQuick precipitation solution (System Biosciences, Palo Alto, CA, USA) according to the manufacturer’s instructions. The EV pellet was retrieved by centrifugation according to the manufacturer’s instructions and was resuspended in phosphate-buffered saline (PBS) for Nanosight tracking analysis (NTA). EVs were visualized on Nanosight NS300 (Malvern Instruments Ltd., Malvern, UK) and the particle size profile and concentration in plasma samples were evaluated with NTA 3.2 analytical software (Malvern Instruments Ltd., Malvern, UK).

For protein extraction, EVs were resuspended in lysis buffer containing RIPA and inhibitors of proteases and phosphatases for protein extraction, as previously described [21]. Briefly, resuspended EVs were sonicated for 2 cycles of 30 secs on ice and incubated on a shaker for 1 h at 4 °C. Proteins from EVs were quantified using the Bradford method (Bio-Rad), according to the manufacturer’s instructions. Ten micrograms of protein was subjected to Western blot analysis for determination of enrichment of exosomal markers (CD9, CD63, CD81, HSP90, diluted at 1:1000 in 5% dry milk in TBS-tween as per the manufacturer’s instructions) in EVs, using the ExoAb antibody kit (SBEXOABKIT1, SBI Systems Biosciences) and normalized using mouse anti-vinculin, followed by goat anti-rabbit or anti-mouse HRP-conjugated secondary antibody.

For ELISA, EV pellets were resuspended in lysis buffer containing PTR 5X Enhancer 50X, inhibitor of proteases (Complete Mini, Roche, Monza, Italy) diluted in deionized water as per the indication of the manufacturer (Abcam). After 20 min incubation at 4 °C, the lysate was stored at 4 °C for the ELISA assays using the Human ACE2 ELISA kit ab235649 and Human TMPRSS2 ELISA kit ab283552 (Abcam).

### 2.6. Immunohistochemistry

Immunohistochemistry was performed on biopsies obtained from 6 NAFLD patients for which formalin-fixed liver sections were available, using the rabbit anti-human ACE2 (ab108252) and rabbit anti-human TMPRSS2 (ab109131, Abcam, Cambridge, UK), both diluted at 1:6400 according to the manufacturer’s instructions, and developed using DAB (3,3′-Diaminobenzidine, Vector Labs) as chromogen as previously described [22]. Histological images were taken using the BX41 microscope (Olympus), coupled with Leica LAS X Life Science software.

### 2.7. Statistical Analysis

Data are reported as mean and standard deviation (SD) or as median and 95% confidence interval (CI) of the median as appropriate. The Pearson or Spearman correlations were performed for continuous normal and not normally distributed variables, respectively. The parametric *t*-test or non-parametric Mann–Whitney test were used as appropriate for continuous variable. A *p*-value less than 0.05 was considered statistically significant. All of the analyses were performed with MedCalc^®^ v.18.9.1 (MedCalc^®^ Software Ltd., Ostend, Belgium).

## 3. Results

### 3.1. Clinical Characteristics of the Study Cohort

Clinical and biochemical characteristics of the study population according to liver etiology are reported in Table 1. The median age of the whole cohort was 56 years (range: 34–86) and 59% of the patients were male. Patients with NAFLD were younger and showed a higher prevalence of obesity and type 2 diabetes compared to those with CHB and CHC. Furthermore, the NAFLD subjects had higher levels of ALT and AST compared to patients with viral etiology, Table 1.

Overall, 13 patients (31%) experienced SARS-Cov-2 infection; specifically, two (15%) had CHB, five (39%) had CHC, and six (46%) had NAFLD. Clinical and biochemical characteristic of the study cohort according to the SARS-Cov-2 infection are reported in Table 2. Among patients who experienced SARS-Cov-2 infection, 85% were males (n = 2 CHB, n = 3 CHC, n = 6 NAFLD); the two females who underwent SARS-Cov-2 infection had CHC. Moreover, SARS-Cov-2 infection was more prevalent in diabetic patients even if the *p*-value did not reach statistical significance, Table 2.

Overall, all patients who experienced COVID-19 disease survived; 3/13 subjects (23.1%) were asymptomatic and only two patients (15.4%) needed hospitalization (one with NAFLD and one with HCV infection). All clinical symptoms are reported in Appendix A. Fever was the most frequent symptom affecting 7/13 (53.8%) patients followed by dyspnoea (30.8%) and coughing (23.1%). Myalgia and arthralgia were the most common symptoms in NAFLD subjects compared to chronic viral hepatitis patients (66.7 vs. 30.8, *p* = 0.0126), Appendix A.

### 3.2. CirculatingACE2 and TMPRSS2 Levels in Chronic Liver Disease Patients

EVs were purified from plasma of all 42 patients and characterized with Nanosight tracking analysis and Western blotting for expression of the classical exosomal markers CD63, CD9, and CD81 described previously [21]. We observed a significantly higher number of EVs in the plasma of NAFLD patients, with respect to CHB and CHC patient plasma (Figure 2a,b). No statistically significant differences were noted in EV counts between NASH and HCV samples.

Analysis of the protein extracted from circulating EVs from the 42 patients divided according to subtypes (NAFLD; CHB and CHC), using ELISA assays, did not reveal detectable ACE2 and TMPRSS2 expression (not shown). On the other hand, plasma levels of ACE2, analyzed by ELISA, were significantly higher in patients with NAFLD compared to those with CHB and CHC (39.7 ± 3.7 ng/mL vs. 10.3 ± 10.8 ng/mL vs. 7.4 ± 3.2 ng/mL, *p* = 0.031), Figure 2c. TMPRSS2 was undetectable in the plasma of all patients (data not shown).

### 3.3. Hepatic Expression of ACE2 and TMPRSS2

In the subgroup of 15 NAFLD patients for whom liver biopsy was available, we explored the hepatic expression of ACE2 and TMPRSS2 by Western blot (native proteins) and IHC analysis. Histological characteristics of the NAFLD cohort are reported in Appendix A.

#### 3.3.1. Western Blot Analysis

Western blotting analysis of ACE2 and TMPRSS2 in Caco-2 cells, used as positive control and human liver samples are depicted in Appendix A, respectively. NAFLD patients who experienced SARS-Cov-2 infection had higher levels of ACE2 compared to those who did not have the infection (3.24 ± 1.49 vs. 1.49 ± 1.32, *p* = 0.0320; Appendix A). On the contrary, hepatic TMPRSS2 levels did not change according to the presence of SARS-Cov-2 infection (2.22 ± 1.10 vs. 1.53 ± 0.91, *p* = 0.2055; Appendix A). Interestingly, NAFLD patients who experienced fever and myalgia due to COVID-19 disease showed a lower expression of TMPRSS2 protein in the liver (3.56 ± 0.21 vs. 1.54 ± 0.43, *p* = 0.0038), Appendix A.

Hepatic ACE2 and TMPRSS2 showed negative correlations with ALT levels (r_S_ = −0.55, *p* = 0.0337 and r_S_ = −0.56, *p* = 0.0351, respectively), Figure 3a,b. Interestingly, we found that hepatic ACE2 and TMPRSS2 were inversely correlated with the amount of steatosis (Figure 3c,d) and were statistically significantly lower in patients with a higher NAS score (Figure 3e,f).

#### 3.3.2. Immunohistochemistry

IHC analysis was performed on 6 out of 15 NAFLD cases, for whom a formalin-fixed liver biopsy section was available. Among these patients, one had cirrhosis, three had advanced fibrosis (F3) and two did not show fibrosis. The IHC analysis revealed an evident increase in ACE2 and TMPRSS2 immune-staining in the cirrhotic patient liver section (Figure 4(1)) compared to less damaged liver tissues (Figure 4(2–6)) and with respect to a control, healthy liver. In the majority of NAFLD patients, ACE2 positivity was detectable in both hepatocytes and and bile ducts, concording with recently reported data [23] while TMPRSS2 positivity was present in both the cytoplasm and nuclei of hepatocytes although the number of positive nuclei differed among different areas of the same biopsy (Figure 4). Specifically, a weak signal seemed to be localized in areas with a lesser amount of hepatic steatosis.

## 4. Discussion

Our data highlight, firstly, that circulating ACE2 is significantly increased in patients with NAFLD compared to those with viral hepatitis. This result may be due to the fact that NAFLD subjects, compared to viral hepatitis patients, are characterized by a continuous state of low grade chronic inflammation that may impact on the liver through the effect of lipids overload. NAFLD is a clinical condition characterized by insulin resistance (IR) mainly in the adipose tissue [24,25]. As a consequence of the impaired lipolysis, triglycerides are continuously hydrolyzed and the flux of free fatty acids from the adipose tissue reach the liver contributing to hepatic steatosis. The lipid overload in the hepatocytes enhances lipo-apoptosis with the consequent release of apoptotic bodies [26].

Secondly, we found that ACE2 and TMPRSS2 expression in the liver was significantly higher in NAFLD patients who experienced SARS-Cov-2 infection (*n* = 6) compared to those who had not undergone viral infection (*n* = 9). This result is coherent with the fact that chronic hepatitis patients with metabolic comorbidities, such as obesity, type 2 diabetes, and hypertension, are more susceptible to viral infection and may potentially develop a more severe COVID-19 disease. This is also supported by the fact that circulating ACE2 is more abundant in the plasma of NAFLD subjects compared to those of chronic viral hepatitis patients. Moreover, despite the elevated expression of these two proteins in the liver, circulating EVs did not contain detectable levels of ACE2 and TMPRSS2, suggesting that EVs are not the principal vehicle for SARS-Cov-2 transport to the liver. However, a more detailed analysis of EV subpopulations, using high-tech methods that can track specifically liver-derived EVs, may be necessary for detecting the presence of ACE2 and TMPRSS2 on an EV’s surface. For instance, El-Shennawy et al. reported an increase in circulating ACE2-expressing CD63^+^ EVs in the plasma of patients with COVID-19, the levels of which are associated with severe pathogenesis, particularly during the acute phase [27]. Interestingly, there was a significantly higher number of EVs in the plasma of NAFLD patients, with respect to CHB and CHC patient plasma. This is in accordance with previous reports of large amounts of EV release upon excessive lipid exposure of hepatocytes [28]. In particular, lipids can enhance EV release by regulating death receptor 5 signaling and ROCK1, through endoplasmic reticulum stress or by mixed lineage kinase 3 [29,30,31].

Finally, we showed that in the liver of patients with NAFLD, the SARS-Cov-2 entry factors ACE2 and TMPRSS2 are inversely correlated with the amount of hepatic steatosis and with the degree of necro-inflammation. These data suggest that ACE2 may have an anti-inflammatory and protective activity in NAFLD subjects who are characterized by low grade chronic inflammation. Several studies demonstrated that ACE2 was able to inhibit liver fibrosis in mice through the degradation of Ang II and the formation of Ang-(1–7). Furthermore, the administration of recombinant ACE2 showed therapeutic effects in models of chronic liver injury and chronic biliary fibrosis [10,32,33]. Even if we did not find any significant association between the hepatic expression of ACE2 and TMPRSS2 with the stages of fibrosis (probably due to the low number of cases), we showed a strong staining for both proteins in liver tissue sections of patients with cirrhosis.

Opposite results were reported in the study by Fondevila et al., where the authors showed a positive correlation between the hepatic expression of ACE2 and TMPRSS with the degree of NAS score. In addition, the authors found a higher expression of SARS-Cov-2 entry factors in NASH patients compared to those with simple steatosis [34]. These data are in contrast to that by Biquard L. and colleagues, in which the authors showed that the expression of ACE2 and TMPRSS2 in the liver was not different in patients with metabolic-associated fatty liver disease even when the patients were stratified by BMI [35]. In our cohort of NAFLD patients, all subjects received the histological diagnosis of NASH based on the joint presence of hepatic steatosis, lobular inflammation, and hepatocytes ballooning. In addition, most of the cases had moderate to severe steatosis (≥33%). When stratified by BMI or type 2 diabetes, we found no differences in the expression of hepatic ACE2 and TMPRSS2, but these results may have been affected by the low number of cases.

This study has some limitations. First of all, NAFLD subjects underwent liver biopsy before the COVID-19 pandemic; however, NAFLD is a clinical condition characterized by a slow evolution. For this reason, among subjects who had an outpatient visit between February and September 2020, we decided to include in the study those who performed a liver biopsy starting from January 2019. Secondly, the small size of the study cohort may affect the interpretation of the results (a borderline statistical significance was often found across the analysis); thirdly, the lack of liver biopsies of CHB and CHC patients did not allow a direct comparison between chronic hepatitis of dysmetabolic and viral etiology. Furthermore, the scarce availability of healthy liver sections also limits the comparison of ACE2 and TMPRSS2 basal levels with those of chronic liver disease tissues.

## 5. Conclusions

Our results indicate that increased ACE2 levels are higher in patients with NAFLD compared to those with viral hepatitis and are more elevated in those who experienced SAR-Cov-2 infection compared to the counterpart. Further analysis including a larger population is necessary to understand why only a subset of patients with pre-existing liver diseases may progress to a worse prognosis following COVID-19 disease.

## Figures and Tables

**Figure 1 viruses-14-02397-f001:**
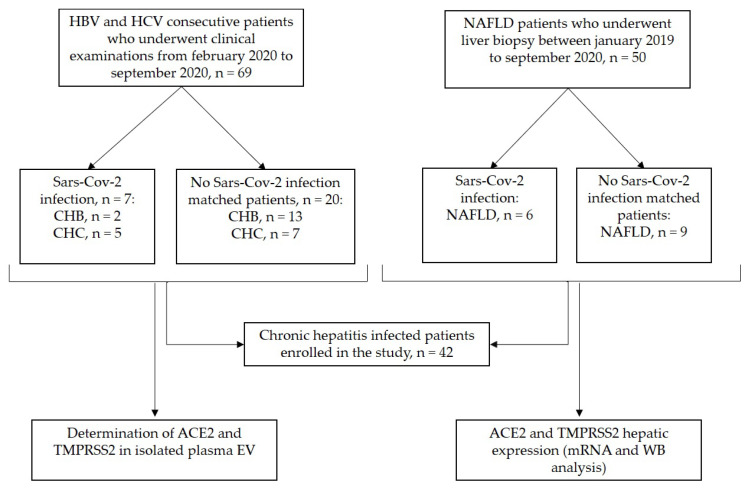
Flow chart of the study. ACE2, angiotensin converting enzyme 2; CHB, chronic hepatitis B; CHC, chronic hepatitis C; NAFLD, non-alcoholic fatty liver disease; SARS-Cov-2, severe acute respiratory coronavirus virus 2; TMPRSS2, transmembrane serine protease 2.

**Figure 2 viruses-14-02397-f002:**
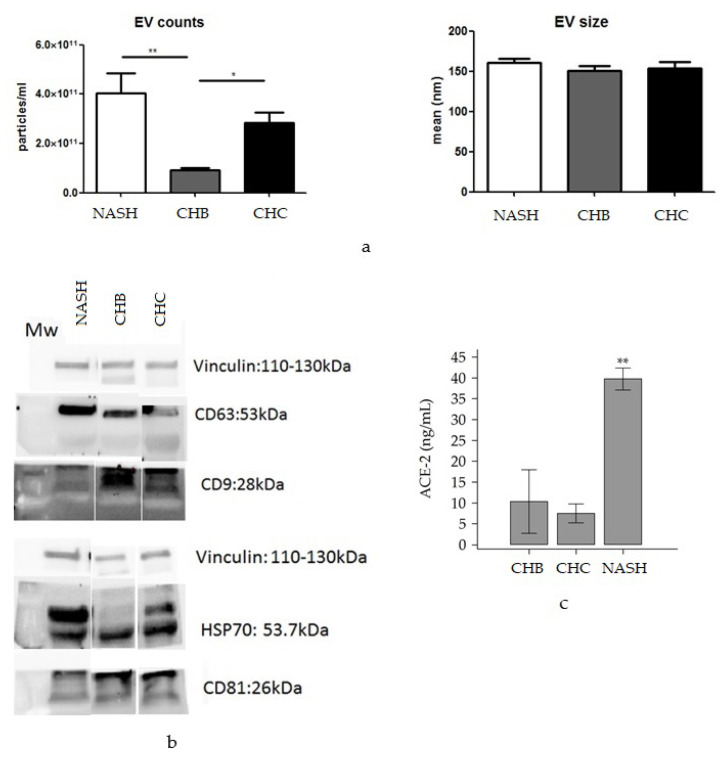
EV counts and dimensional distribution of hepatic EV according to etiology (**a**). Western blot analysis of EV markers CD63, CD9, HSP70 and CD81 expression, normalized according to Vinculin expression (**b**). EV, extravesicles; MW, molecular weight marker; NAFLD, non-alcoholic fatty liver disease. (**c**) Circulating levels of ACE2 in plasma of CHB, CHC and NAFLD patients. The image shows the circulating ACE2 levels (ng/mL) in the plasma of patients with liver disease of different etiology. Circulating ACE2 was significantly higher in NAFLD subjects compared to CHB and CHC patients (*p* = 0.0019). * *p* < 0.05; ** *p* < 0.005.

**Figure 3 viruses-14-02397-f003:**
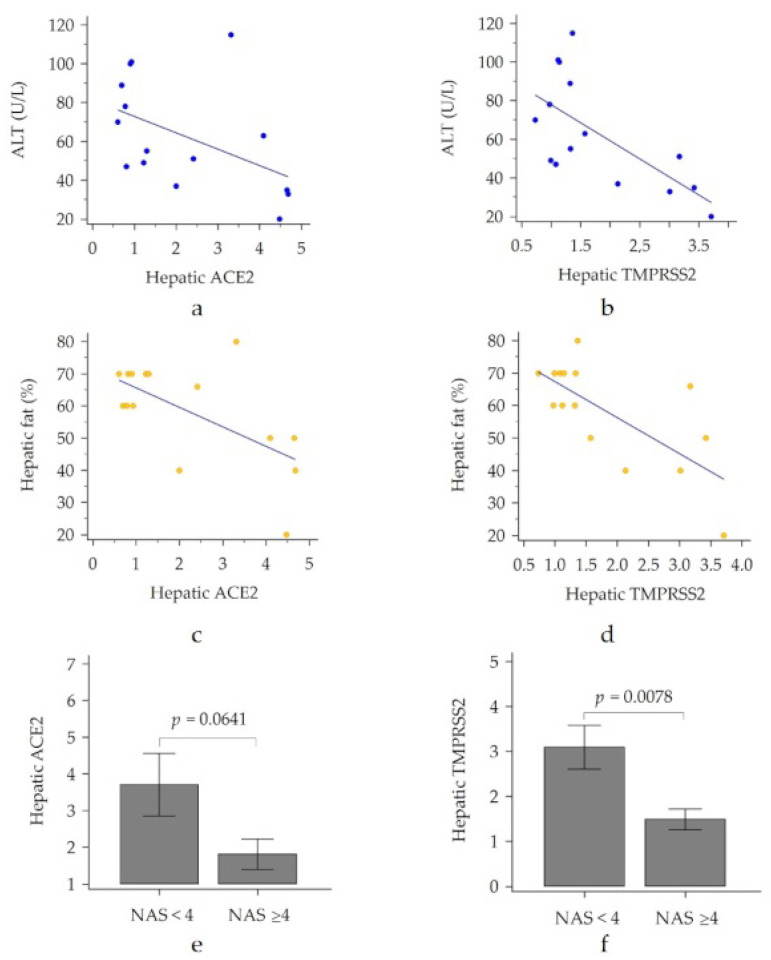
Hepatic ACE2 and TMPRSS2 according to the amount of steatosis (**a**–**d**) and to the degree of NAS score (**e**,**f**). ACE2, angiotensin converting enzyme 2; NAS, non-alcoholic fatty liver disease activity score; TMPRSS2, transmembrane serine pro-tease 2.

**Figure 4 viruses-14-02397-f004:**
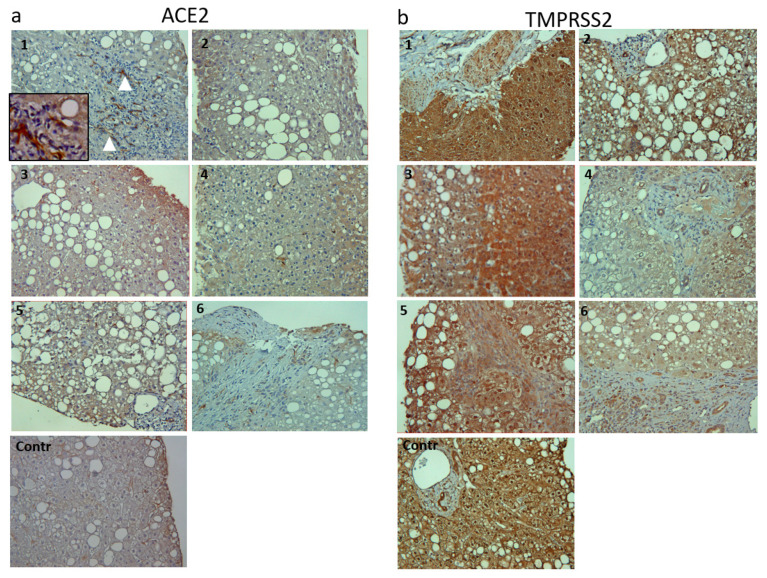
Immunohistochemistry staining for ACE2 (**a**), and TMPRSS2 (**b**), images have been taken at 10× magnification. White arrowheads show ACE2 expression in a cirrhotic patient liver. Inset at 20× magnification highlights ACE2-positivity (brown). ACE2, angiotensin converting enzyme 2; Contr, control; TMPRSS2, transmembrane serine pro-tease 2. Cases 1 = Cirrhosis; Cases 2,3 = F0; Cases 4–6 = F3.

**Table 1 viruses-14-02397-t001:** Clinical and biochemical characteristics of the study cohort according to the liver disease etiology.

Variables	All *n* = 42	CHB *n* = 15	CHC *n* = 12	NAFLD *n* = 15	*p*-Value
Age, years	56 (53–62)	56 (52–63)	60 (54–67)	53 (44–60)	0.2847
Male gender, n (%)	25 (59.5%)	9 (60%)	5 (41.7%)	11 (73.3%)	0.2495
BMI, kg/m^2^	26.8 (25.5–29.3)	25.7 (24.3–26.8)	24.9 (23.3–26.6)	30.9 (29.6–32.0)	<0.0001
Obesity, n (%)	15 (35.7%)	3 (20%)	1 (8.3%)	11 (73.3%)	0.0006
Type 2 diabetes, n (%)	11 (26.2%)	0 (-)	2 (16.7%)	9 (60%)	0.0006
AST, U/L	25 (22–29)	22 (20–27)	24 (17–27)	37 (26–47)	0.0183
ALT, U/L	26 (20–37)	23 (20–26)	14 (12–25)	55 (40–86)	<0.0001
Platelets, ×10^9^/L	213 (200–251)	200 (186–242)	205 (166–244)	273 (225–282)	0.0279

Data are reported as median and 95% confidence intervale (CI) of the median or as frequency (n) and percentage (%). ALT, alanine aminotransferase; AST, aspartete aminotransferase; BMI, body mass index; CHB, chronic hepatitis B; CHC, chronic hepatitis C; NAFLD, non-alcoholic fatty liver disease.

**Table 2 viruses-14-02397-t002:** Clinical and biochemical characteristics of the study cohort according to the SARS-Cov-2 infection.

Variables	No Infection *n* = 29	SARS-Cov-2 Infection, *n* = 13	*p*-Value
Age, years	53 (47–58)	51 (39–65)	0.9060
Male gender, n (%)	14 (48%)	11 (85%)	0.0284
Etiology, n (%) CHB	9 (31.1%)	2 (15.4%)	0.6672
CHC	7 (24.1%)	5 (38.5%)	0.6077
NAFLD	13 (44.8%)	6 (46.1%)	0.9589
BMI, kg/m^2^	25.7 (24.5–27.5)	29.4 (26.8–31.5)	0.0702
Obesity, n (%)	8 (27.6%)	7 (53.8%)	0.1047
Type 2 diabetes, n (%)	60 (20.7%)	5 (38.5%)	0.0516
AST, U/L	26 (22–29)	25 (18–41)	0.7032
ALT, U/L	24 (20–39)	35 (18–64)	0.4540
Platelets, ×10^9^/L	218 (199–258)	209 (181–281)	0.9566

Data are reported as median and 95% confidence intervale (CI) of the median or as frequency (n) and percentage (%). ALT, alanine aminotransferase; AST, aspartete aminotransferase; BMI, body mass index; CHB, chronic hepatitis B; CHC, chronic hepatitis C; NAFLD, non-alcoholic fatty liver disease.

## Data Availability

The data presented in this study are available upon request from the corresponding author.

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
