# Peer review of "Expression of SARS-Cov-2 Entry Factors in Patients with Chronic Hepatitis"

_viruses, 2022, doi:10.3390/v14112397_

Round 1

Reviewer 1 Report

The issue addressed in this report is the role of of the infection by SARS-CoV-2 on the severity of liver disease (patients with intact kidneys).

The investigators found that circulating ACE2 are greater in patients with NAFLD in comparison with patients showing chronic viral hepatitis (P<0.001).

According to Western blotting analysis, the levels in the liver of ACE2 were higher in patients with NAFLD  and SARS-CoV-2 infection than those patients with NAFLD without SARS-CoV-2 infection (P<0.03). 

On the contrary, the hepatic levels of TMPRSS2 did not change according to the presence of SARS-CoV-2 infection. 

Of note, an inverse relationship occurred between liver levels of ACE2 and TMPRSS2 and the amount of hepatic steatosis. This is in clear contrast with the data reported in the medical literature and appropriately cited by the authors in the Section Discussion of the manuscript (Fondevila M, et al. Biquard L, et al)

The issue addressed by Rosso et al. is very hot - however, the number analysed are very small. I do believe that the number of patients with chronic hepatitis on follow-up in Torino University Hospital is much higher than the number of patients enrolled in the study (n=42 unique patients). As an example, western blot analysis was made in the liver tissue of a subgroup of only 15 patients. This is an important shortcoming of the study. I feel, however, that research on this issue is expensive, laborious and time-consuming. On the other side, the results reported here can prompt additional investigations. 

Author Response

R: We thank the reviewer for his/her comment and we agree with his/her observations. Published data showed conflicting results about this topic and further studies are needed to elucidate mechanisms of SARS-Cov-2 infection in different districts, including the liver. We are aware that the low number of patients may affect the interpretations of the results. The main difficulty was related to the enrolment of patients due to the reduced and delayed outpatient activities during pandemic period. Nevertheless, we think that our preliminary results may be a starting point for further investigations.

Reviewer 2 Report

In the paper named “Expression of SARS-Cov-2 entry factors in patients with chronic viral hepatitis” authors explore the impact of SARS-Cov-2 infection on disease severity in a group of patients with Chronic hepatitis.  They found that NAFLD subjects have higher circulating ACE2 levels. Similarly, hepatic expression of ACE2 was higher in subjects who underwent SARS-Cov2 infection compared to the counterpart. On the other hand author found hepatic TMPRSS2 was significantly lower in patients who experienced symptomatic COVID-19 disease compared to asymptomatic patients. Moreover authors indicate that further studies are necessary to understand the impact of COVID-19 in patients with pre-existing liver diseases.

Only minor points are required

1)      In figure 1 the n of patients HBV and HCV  enrolled n=69 do not mach with the number indicate with plus without SARS-Cov -2 infection (total n=27) the same happens in NAFLD patients

2)      In figure 1 author give data from HBV and HCV but perhaps the better nomenclature is CHB and CHV

3)      The paragraph between line 83 and 85 is confusing please clarify

4)      In point 2.3 data about the antibody dilution is missing

5)      In the same point 2.3 author use RNA later to obtain protein have they tested the protein quality??

6)       How they make the EVs lyses? This data is missing in point 2.5

7)      In point 2.5 the exosomal markers dilution is missing

8)      In figure 2 I think that the samples must be named as CHB and CHV not by the virus name. In EV Counts are not significative differences between NASH and HCV?

9)      The paragraph between lines 207-208 is part of the discussion

Author Response

R: We thank the Reviewer for his/her helpful comments, which have helped us to improve our manuscript. Please find the answers to each comment below.

Specific comments

  • In figure 1 the n of patients HBV and HCV enrolled n=69 do not mach with the number indicate with plus without SARS-Cov -2 infection (total n=27) the same happens in NAFLD patients

R:  We thank the reviewer for the comment. As explained at the beginning of the paragraph 2.1 “Study population”, from 69 consecutive patients with viral hepatitis and 50 consecutive patients with NAFLD, only 7 and 6 patients, respectively, accepted to participate in the study (line 70-75). To avoid misinterpretation, we revised the flow chart accordingly.

  • In figure 1 author give data from HBV and HCV but perhaps the better nomenclature is CHB and CHV

R: We have changed the nomenclature in Figure 1 accordingly.

  • The paragraph between line 83 and 85 is confusing please clarify

R: We agree with the reviewer and we have rephrased the sentence into: “Overall, out of the 42 patients enrolled, 15 patients (35.7%) had chronic hepatitis B [CHB] infection, 12 (28.6%) had chronic hepatitis C [CHC] infection and 15 (35.7%) suffered from non-alcoholic fatty liver diseases [NAFLD]”, (line 85-87).

  • In point 2.3 data about the antibody dilution is missing

R: We have added the antibody dilutions in the text (line 115-119).

  • In the same point 2.3 author use RNA later to obtain protein have they tested the protein quality??

R: The protein quality was confirmed by Ponceau S (Ponceau S stain (0.01% (w/v) Ponceau S in 1% acetic acid (v/v))) staining after Sodium Dodecyl Sulfate Polyacrylamide Gel Electrophoresis (SDS–PAGE) loading and membrane transfer as previously described (doi: 10.1016/j.ab.2019.03.010). No smears, representative of degraded proteins, were observed (see the figure below). Thus, protein integrity was conserved with the extraction method used. Please see blot enclosed below. A sentence has been added in the methods section accordingly. The protein concentration varied from 2.9 to 25.6 mg/ml depending on the starting size of the biopsies. 

  • How they make the EVs lyses? This data is missing in point 2.5

R: We have added in the text details regarding the EV lysis (line 135-149).

  • In point 2.5 the exosomal markers dilution is missing

R: This information has been added in the text (line 153).

  • In figure 2 I think that the samples must be named as CHB and CHV not by the virus name. In EV Counts are not significative differences between NASH and HCV?

R: We have changed the nomenclature accordingly. Concerning EV counts, no statistically significant differences were noted in NASH vs. HCV samples. This information has been added in the text (line 218-219).

  • The paragraph between lines 207-208 is part of the discussion

R: We agree with the reviewer and we shifted this part in the discussion section (305-310).
